# Relevance of near-surface soil moisture vs. terrestrial water storage for global vegetation functioning

Prajwal Khanal[1,2,3], Anne J. Hoek Van Dijke[1], Timo Schaffhauser[2], Wantong Li[1], Sinikka J. Paulus[1,4,5], Chunhui Zhan[1,6], René Orth[1,5]

[1]Department of Biogeochemical Integration, Max Planck Institute for Biogeochemistry, Hans-Knöll-Straße 10, 07745 Jena, Germany
[2]Chair of Hydrology and River Basin Management, Technical, University of Munich, Arcisstraße 21, 80333 Munich, Germany
[3]Faculty of Geo-Information Science and Earth Observation (ITC), University of Twente, Langezijds, 7500 AE, Enschede, The Netherlands
[4]Chair of Terrestrial Ecohydrology, University of Jena, Burgweg 11, 07749 Jena, Germany
[5]Chair of Modeling of Biogeochemical Systems, Faculty of Environment and Natural Resources, University of Freiburg, Tennenbacher Straße 4, 79106 Freiburg, Germany
[6]Chair of Land Surface-Atmosphere Interactions, Technical University of Munich, TUM School of Life Sciences Weihenstephan, 85354 Freising, Germany

*Correspondence to*: Prajwal Khanal (ktm.prajwalkhanal@gmail.com)

**Abstract.** Soil water availability is an essential prerequisite for vegetation functioning. Vegetation takes up water from varying soil depths depending on the characteristics of their rooting system and soil moisture availability across depth. The depth of vegetation water uptake is largely unknown across large spatial scales as a consequence of sparse ground measurements. At the same time, emerging satellite-derived observations of vegetation functioning, surface soil moisture and terrestrial water storage, present an opportunity to assess the depth of vegetation water uptake globally. In this study, we characterise vegetation functioning through the Near-Infrared Reflectance of Vegetation (NIRv) and compare its relation to (i) near-surface soil moisture from ESA-CCI and (ii) total water storage from GRACE at the monthly time scale during the growing season. The relationships are quantified through partial correlations to mitigate the influence of confounding factors such as energy and other water-related variables. We find that vegetation functioning is generally more strongly related to near-surface soil moisture, particularly in semi-arid regions and areas with low tree cover. In contrast, in regions with high tree cover and in arid regions, the correlation with terrestrial water storage is comparable to or even higher than with near-surface soil moisture, indicating that trees can and do make use of their deeper rooting systems to access deeper soil moisture, similar to vegetation in arid regions. At the same time we note that this comparison is hampered by different noise levels in these satellite data streams. In line with this, an attribution analysis that examines the relative importance of these soil water storages for vegetation reveals that they are controlled by (i) water availability influenced by the climate and (ii) vegetation type reflecting adaptation of ecosystems to local water resources. Next to variations in space, the vegetation water uptake depth also varies in time. During dry periods, the relative importance of terrestrial water storage increases, highlighting the relevance of deeper water

resources during rain-scarce periods. Overall, the synergistic exploitation of state-of-the-art satellite data products to disentangle the relevance of near-surface vs. terrestrial water storage for vegetation functioning can inform the representation of vegetation-water interactions in land surface models to support more accurate climate change projections.

## 1. Introduction

The regulation of water, energy, and biogeochemical cycling between land and atmosphere is primarily dependent on vegetation. In addition, global vegetation provides essential ecosystem services such as food production and uptake of some of the anthropogenic carbon dioxide emissions (Keenan and Williams, 2018). Vegetation growth depends on nutrient, water and energy availability. As a result, on a global scale, there are regions with energy or water limited vegetation functioning (Orth, 2021). In energy-limited regions, the functioning of vegetation is controlled by radiation and temperature, as they often lack sunny and warm conditions but have ample soil moisture. In contrast, soil moisture becomes critical for vegetation growth in water-limited regions. Plant photosynthesis involves opening the stomata for the uptake of $CO_2$, while at the same time water is lost through transpiration. However, in water-limited conditions, plants can reduce the stomatal opening to avoid water loss, leading to a decrease in photosynthesis. Hence, variations in soil moisture are likely to affect vegetation functioning in water-limited conditions. Moreover, climate change has led to an expanded water limitation on vegetation (Denissen et al., 2022) and increased vegetation sensitivity to soil moisture (Li et al., 2022). For these reasons, it is essential to better understand the dependence of vegetation functioning on soil moisture to comprehend their coping mechanisms during drought to predict the future of global water, energy, and carbon cycles.

Plants extract water from varying soil depths based on the positioning of their roots and the availability of soil moisture and nutrients. In general, the plant water uptake depth further differs spatially across different climate regimes and vegetation types, and temporally between seasons. Vegetation in arid regions is more susceptible to fluctuations in near-surface soil moisture compared to vegetation in humid regions (Xie et al., 2019). Grasses, which generally have shorter roots than trees and shrubs, are more reliant on near-surface moisture than deeper moisture (Schenk and Jackson, 2002). Further, root water uptake profiles vary within individual plant types according to above-ground biomass and age, with larger and older trees having deeper roots capable of extracting water from deeper soil layers (Schenk and Jackson, 2002; Tao et al., 2021). Additionally, within similar climate regimes, plant water uptake varies across topographic positions. Upland and lowland roots tend to be shallower, making vegetation more reliant on near-surface soil moisture, while roots go deeper in steep terrain between these landscapes to access both surface and deep moisture (Fan et al., 2017).

Though spatial variations of plant water uptake depths across vegetation types and climate regimes, and temporal shift during dry-months, are widely studied at point scale, inadequate deep soil moisture records pose a major obstacle to study vegetation root water uptake at a global scale. Microwave remote sensing allows to infer near-surface soil moisture dynamics globally.

While microwaves penetrate only the top few centimeters and do not cover the entire soil moisture profile, they represent larger depths of moisture variation, providing valuable insights into at least some of the root zone soil moisture (Feldman et al., 2023). Land surface models provide an alternative source of global soil moisture data across depths, but they are subject to uncertainties arising from meteorological data, inaccurate knowledge of soil and vegetation characteristics, and the representation of complex processes such as photosynthesis, infiltration, and evaporation (Koster et al., 2009; Seneviratne et al., 2010). Hence, some studies have employed reanalysis-based soil moisture estimates, to investigate the relationship between vegetation and soil moisture at the global scale (Li et al., 2021; Miguez-Macho and Fan, 2021); but those are likely to be impacted by model assumptions affecting soil moisture dynamics, particularly for deeper layers where less observational constraints are available. Thus, studying vegetation interactions with the entire water column, including near-surface and deep soil moisture, at a global scale using exclusively observation-based dataset is imperative to enhance the understanding of relevance of near-surface and deep soil moisture for vegetation functioning.

The Gravity Recovery and Climate Experiment (GRACE) satellite mission, launched in 2002, provides total water storage (TWS) anomalies observations at the global scale. The TWS captures not only soil water but also snow and ice, canopy water, surface water and groundwater. Its depth of representation is therefore difficult to physically quantify, and that is why we study TWS anomalies. Nevertheless, they seem to be related to variations of overall water availability (near-surface + deep soil moisture) for vegetation (Yang et al., 2014). The inter-annual carbon dioxide growth rate in the atmosphere, for example, has been found to be well correlated with the total water storage anomalies on a global scale, indicating the relevance of total water column for vegetation functioning (Humphrey et al., 2018). In this study, we assume that TWS anomalies can be used to estimate the variation of overall water availability (near-surface + deep soil moisture) for vegetation under (i) snow-free conditions, and assuming that (ii) water storage variations in lakes or groundwater are negligible at the monthly time scale, (iii) and canopy water storage is much smaller than soil water storage and hence also negligible (Zheng and Jia, 2020; Stocker et al., 2023). While soil moisture fluctuations represent the largest variation of TWS (Rodell and Famiglietti, 2001), it is essential to note that certain regions exhibit notable short term fluctuations in lake and groundwater due to human management (Strassberg et al., 2007; Cooley et al., 2021).

This study focuses on understanding the relevance of near-surface soil moisture vs. total water storage for vegetation functioning on a global scale using observation-based datasets, thereby inferring vegetation's large-scale water uptake depth from observation-based datasets. For this purpose, we utilise TWS and near-surface soil moisture and correlate them with vegetation functioning, represented by Near-Infrared Reflectance of Vegetation (NIRv). In particular, we analyse (1) what is the relevance of near-surface soil moisture vs. the terrestrial water storage for vegetation functioning?, (2) how does the importance of near-surface soil moisture vs. terrestrial water storage change during dry months? and (3) how do climatic, vegetation, and topographic characteristics explain the variability in the relevance of near-surface vs. terrestrial water storage for vegetation functioning?

## 2. Data and Methodology

**Table 1: Table summarising all the datasets.**

| Datasets | Variables | Source | Spatial Resolution | Temporal Resolution | Temporal Coverage | References |
|---|---|---|---|---|---|---|
| **Vegetation Functioning** | Near Infrared Reflectance of Vegetation (NIRv) | MODIS/MOD13C1 v061 | 0.05 degree | 16 daily | 2000 - present | (Badgley et al., 2017) |
| | Solar Induced Chlorophyll Fluorescence (SIF) | GOME-2 | 0.5 degree | 16 daily | 2007 - 2018 | (Köhler et al., 2015) |
| **Soil Water Storage** | Near-surface soil moisture (SSM) | ESA-CCI v04.4 | 0.25 degree | Daily | 1978 - 2022 | (Dorigo et al., 2017) |
| | Total Water Storage (TWS) Anomalies | GRACE | 0.5 degree | Monthly | 2002 - present | (Landerer and Swenson, 2012) |
| **Meteorological** | Air Temperature ($T_a$) | ERA-5 | 0.25degree | Hourly | 1940 - present | (Hersbach et al., 2020) |
| | Precipitation (P) | | | | | |
| | Net Radiation ($R_n$) | | | | | |
| | Dew point Temperature ($T_d$) | | | | | |
| **Climatological** | Aridity Index | Global Aridity Index and Potential Evapotransp | 30 arc seconds | Static | 1970-2000 | (Zomer et al., 2022) |

| | | iration Database - Version 3 | | | | |
|---|---|---|---|---|---|---|
| **Vegetation and Land cover class** | Tree cover fraction | VFC5KYR | 0.05 degree | | 1982 - 2016 | (Hansen, Matthew and Song, Xiao-Peng, 2018) |
| | Land cover data | ESA-CCI | 300 m | Yearly | 1992 - 2018 | ESA. Land Cover CCI Product User Guide Version 2. Tech. Rep. (2017) |
| **Topographical data** | Elevation | Earthenv | 1 km | Static | | (Amatulli et al., 2018) |
| | Slope | | | | | |
| **Soil data** | Fraction of sand | FAO | 0.05 degree | Static | | (Reynolds et al., 2000) |
| | Fraction of clay | | | | | |
| **Irrigation** | Percentage of Irrigated area | HID | 5 arcmin | Yearly | 1990 - 2005 | (Siebert et al., 2015) |

101

**2.1 Data**

**2.1.1 Vegetation Functioning:**

In our study, vegetation functioning is characterised by satellite measurements of Near-Infrared Reflectance of vegetation (NIRv) and Solar Induced Fluorescence (SIF) (**Table 1**). NIRv is the product of near-infrared reflectance and the normalised difference vegetation index (NDVI) and represents the vegetation structure and vegetation greenness (Badgley et al., 2017). The NIRv data is available at a high spatial resolution of 0.05°, and the original 16-day data was aggregated to the monthly NIRv data. SIF is directly related to the photosynthetic activity of plants because the excess energy from sunlight, that triggers the light reaction during photosynthesis, is dissipated by leaf as chlorophyll fluorescence (Mohammed et al., 2019). SIF data is derived from the Global Ozone Monitoring Experiment (GOME-2), because GOME-2 provides relatively reliable data over

a long period (2007-2018). The 0.5° spatial and 16-day temporal resolution SIF data is processed into monthly data as described by (Köhler et al., 2015).

The high spatial resolution of NIRv allows for a detailed study of the correlation of vegetation functioning with soil water availability. Therefore, we performed the main analyses using NIRv data. However, SIF is more sensitive to drought stress than NIRv (Qiu et al., 2022). Therefore, we perform additional analyses with SIF to show that the relationships hold for a different and more direct indicator of vegetation functioning.

**2.1.2 Soil Water Storage**

This study includes two different measures of soil water availability. The near-surface soil moisture (SSM) provides an estimate of water availability in the top layer of the soil, while the Terrestrial Water Storage (TWS) Anomaly provides an estimate of the overall water column of the soil. The SSM data is derived from the European Space Agency (ESA) Climate Change Initiative Program (CCI), which combines active and passive satellite microwave measurements to provide reliable estimates of SSM (Dorigo et al., 2017). The ESA CCI soil moisture data, at a daily temporal resolution, was aggregated to monthly temporal resolution. The TWS Anomaly data is derived from the GRACE mission, which measures changes in the Earth's gravity field (Landerer and Swenson, 2012). Here, we use the JPL-Mascons product of TWS Anomalies which is available at a 0.5° spatial and monthly temporal resolution (Watkins et al., 2015).

**2.1.3 Meteorological Data**

Employed climate variables include monthly air temperature($T_a$), 2m dew point temperature ($T_d$), precipitation (P), and net radiation ($R_n$) from the ERA5 reanalysis products at a 0.25° spatial resolution. The vapor pressure deficit (vpd) is calculated from $T_a$ and $T_d$. Further, the aridity index is calculated from the ratio between the long-term mean $R_n$ (mm $y^{-1}$) (1 MJ/sq.m/day = 0.408 mm/day) and P (mm $y^{-1}$) for each grid cell (Budyko, 1974). We opted for this formulation as it offers a direct estimation of aridity and water (energy) constraints on vegetation. This eliminates the necessity to navigate through various formulations utilized for calculating potential evapotranspiration. However, we conducted additional validations of our results using the Global Aridity Index dataset (Zomer et al., 2022) based upon the FAO Penman-Monteith Reference Evapotranspiration equation. The use of the Global Aridity Index did not change the results of our study (**Section 3.4**). In addition, the mean and standard deviation of the climate variables are calculated and incorporated in the attribution analysis (**Section 2.2.3**).

**2.1.4 Vegetation, soil, and topography data**

To evaluate the resulting correlation of vegetation functioning and water storages with respect to vegetation characteristics, we employ the tree cover fraction data from the AVHRR vegetation continuous fields products (VCF5KYR, https://lpdaac.usgs.gov/products/vcf5kyrv001/) (Hansen, Matthew and Song, Xiao-Peng, 2018). For this purpose, the mean of tree cover fraction for the years between 2007 and 2016 is calculated.

Topographical variables such as elevation and slope are incorporated along with other meteorological variables to determine the relative contribution of different variables to the correlation between vegetation functioning and water storage. Topographic data at a 5 km resolution were downloaded from the EarthEnv. These data are calculated based on the 250 m GMTED dataset and compared against the 90 m SRTM 4.1 dev dataset. The data were resampled to a coarser resolution of 5 km using various aggregation techniques, details of which are in Amatulli et al., 2018. Furthermore, for each grid cell, the fraction of sand and clay in soil (Reynolds et al., 2000) along with the percentage of irrigated area (Siebert et al., 2015) were considered in attribution analysis.

## 2.2 Methodology

### 2.2.1 Data pre-processing

A flowchart of the data pre-processing and analyses is presented in **Figure S1**. The time period of analysis is from 2007 to 2018 constrained by the concurrent availability of all involved datasets. All the analyses were performed in monthly temporal resolution and at 0.05° spatial resolution (for NIRv) and 0.5° spatial resolution (for SIF). The SSM and TWS data were initially available at 0.25° and 0.5° resolution, but were disaggregated or aggregated to 0.05° or 0.5° degrees, depending on the spatial resolution of the analysis performed, based on the assumption that the soil water storage anomalies are representative over larger areas. Also, the meteorological data and vegetation, soil, and topographic data were resampled into the same resolution. After aggregating all the datasets to 0.05° resolution, the monthly anomalies were calculated by subtracting the long term mean monthly cycle and by removing linear trends. A SIF threshold was applied in each grid cell to filter out non-growing season data. For this purpose, we filtered out all the months from 2007-2018 when the mean-monthly SIF value was below the threshold of 0.2 mW/m2/sr/nm. We apply an additional temperature threshold ($T_a > 5$℃) to remove the months with frozen soil and snow cover, similar to (Li et al., 2021). Last, all months with missing soil water storage or vegetation functioning records were excluded.

### 2.2.2 Calculate the relevance of near-surface (SSM) soil moisture and terrestrial water storage (TWS) for vegetation functioning

We calculated the Spearman correlation between vegetation functioning (NIRv) and soil water storages (SSM and TWS) for each grid cell during growing season months when observations for at least 40 months were available. To derive partial correlation estimates between NIRv and the water storages, we employed a bootstrapping approach (resampling with replacement from the original data) within each grid cell, with 1000 repetitions to compute bootstrap means and confidence intervals. The cutoff of 40 months was implemented to guarantee a substantial number of observations for growing-season months in each grid cell. This consideration assumes that the minimum number of growing-season months varies from 3 to 4 months per year globally. In addition to soil moisture, also air temperature ($T_a$) and net radiation ($R_n$) affect the vegetation functioning. Moreover, SSM (soil moisture) and TWS (total water storage) demonstrate a notable correlation, as illustrated in

**Figure S2**, signifying the presence of mutual information. To exclusively examine the individual impacts of each water storage

variable on vegetation functioning and disentangle mutual information from other water variables, we accounted for

confounding effects. This entailed computing the partial correlation between NIRv and water storages (SSM or TWS), while

controlling for Ta, Rn, and the other water storage variable (TWS or SSM). Since we focus on understanding the role of soil

moisture on vegetation functioning, which is primarily critical in water-limited conditions, we removed the grids cells with

negative partial correlations from our analysis. Such negative partial correlations may hint at vegetation's converse effect on

soil moisture (when increasing vegetation activity depletes the soil moisture) and a negative correlation could occur in the grid

cells where water limits vegetation productivity through oxygen limitation (Ohta et al., 2014). Also, note that predominant

energy limitation of the vegetation prevents the evaluation of the relevance of soil moisture vs. terrestrial water storage as

partial correlations will become insignificant when temperature or radiation are mainly controlling vegetation functioning.

It is important to note that we chose not to apply a significance criterion in analyzing the partial correlation between NIRv and

water storages. When controlling for both water storage (TWS or SSM) and energy variables (Ta and Rn) in the partial

correlation (NIRv~SSM or TWS), a limited number of grid cells demonstrate significant correlation globally, given the high

correlation between SSM and TWS (**Figure S2**). This poses challenges for drawing global inferences on vegetation water

uptake. However, our overarching goal is to discern variations in the partial correlation of NIRv with water storages across

differing climate-vegetation gradients and how it changes from the growing season to dry months, rather than confirming

specific statistical thresholds. For this, we want to maintain a sufficient amount of grid cells necessary for making global

inferences. However, to ensure that our results are not affected by the significance criterion, we conducted additional analyses

considering only grid cells with a significant partial correlation (though a very small number compared to the total grid cells

available for each AI-TC class globally), as described in **section 3.4**.

The impact of all pre-processing steps on the number of grid cells included in this study is illustrated in **Figure S3**. Generally,

our filtering procedures enable us to concentrate primarily on water-limited regions, as they effectively remove a substantial

number of grid cells from the wet regions globally.

To analyse how the importance of SSM and TWS changes during dry months, we specifically selected the months characterized

by the lowest 10% SSM for each grid cell, representing the driest conditions within the growing-season months. The partial

correlations between NIRv and water storages, r(NIRv~SSM) and r(NIRv~TWS) were calculated separately for dry months.

To focus on vegetation response to similar extent of dryness spatially, only grid cells with greater than 100 monthly

observations were considered for the dry months analysis. In addition, only the grid cells which had positive partial correlation

in growing season months were included for the dry months analysis.

After computing the partial correlations, we grouped the grid cells by aridity and tree cover classes, which allowed us to analyse the evolution of correlations and the difference between the partial correlation across aridity-tree cover classes. Afterwards, we again employed a full bootstrapping methodology with 1000 repetitions to compute the bootstrap means and confidence interval for each aridity-tree cover class with sufficient number of observations for both growing season and dry months. Moreover, to test the robustness of the results, we did additional partial correlation analyses, for which we correlated the SIF (instead of NIRv) with SSM and TWS. The analyses with SIF were performed at a spatial resolution of 0.5°, at which SIF data was available.

### 2.2.3 Attribution Analysis

We used a random forest model to understand the spatial variability in the relevance of SSM versus TWS for NIRv. Random forest is a nonparametric based regression algorithm which does not require any statistical assumptions on the predictor and target variables which makes it particularly useful for detecting the nonlinear relationship (Breiman, 2001). Given potential nonlinear impacts of various factors (climate, soil types, vegetation) on the relationship between moisture storages and vegetation functioning, this study employed the random forest method to assess the relative contributions of these variables.

In our study, 15 predictors were included in the random forest model based on their potential physical relevance to the target variable, which is the difference in correlation between SSM and TWS with NIRv in growing season months. These predictors included mean and standard deviation of climate variables ($T_a$, $R_n$, P and vpd), aridity index, topographical variables (elevation and slope), vegetation variable (tree cover), soil-related variables (fraction of clay and sand), and percentage of irrigated areas for each grid cell. We calculated the mean and standard deviation of the climate variables only during the growing-season months, as determined for the subsequent partial correlation analysis. Furthermore, only the grid cells exhibiting positive partial correlation between NIRv and SSM as well as NIRv and TWS during growing season-months were included in the random forest analysis. For training a random forest model, we used the "xgboost" package in R (Chen and Guestrin, 2016).

We further incorporate SHAP (SHapley Additive exPlanations) values for interpreting the predictions of the random forest model (Lundberg et al., 2020). The SHAP value for a feature is the average difference in prediction of the model when that feature is included compared to when it is excluded, over all possible combinations of features. By calculating SHAP values for each feature in the model, we identified which features were most important in explaining the spatial variability in the relevance of SSM versus TWS. For calculating the SHAP values, we employed "SHAPforxgboost" package in R.

**3. Results and Discussion**

**3.1 Coupling of vegetation functioning with surface soil moisture and total water storage in the growing season**

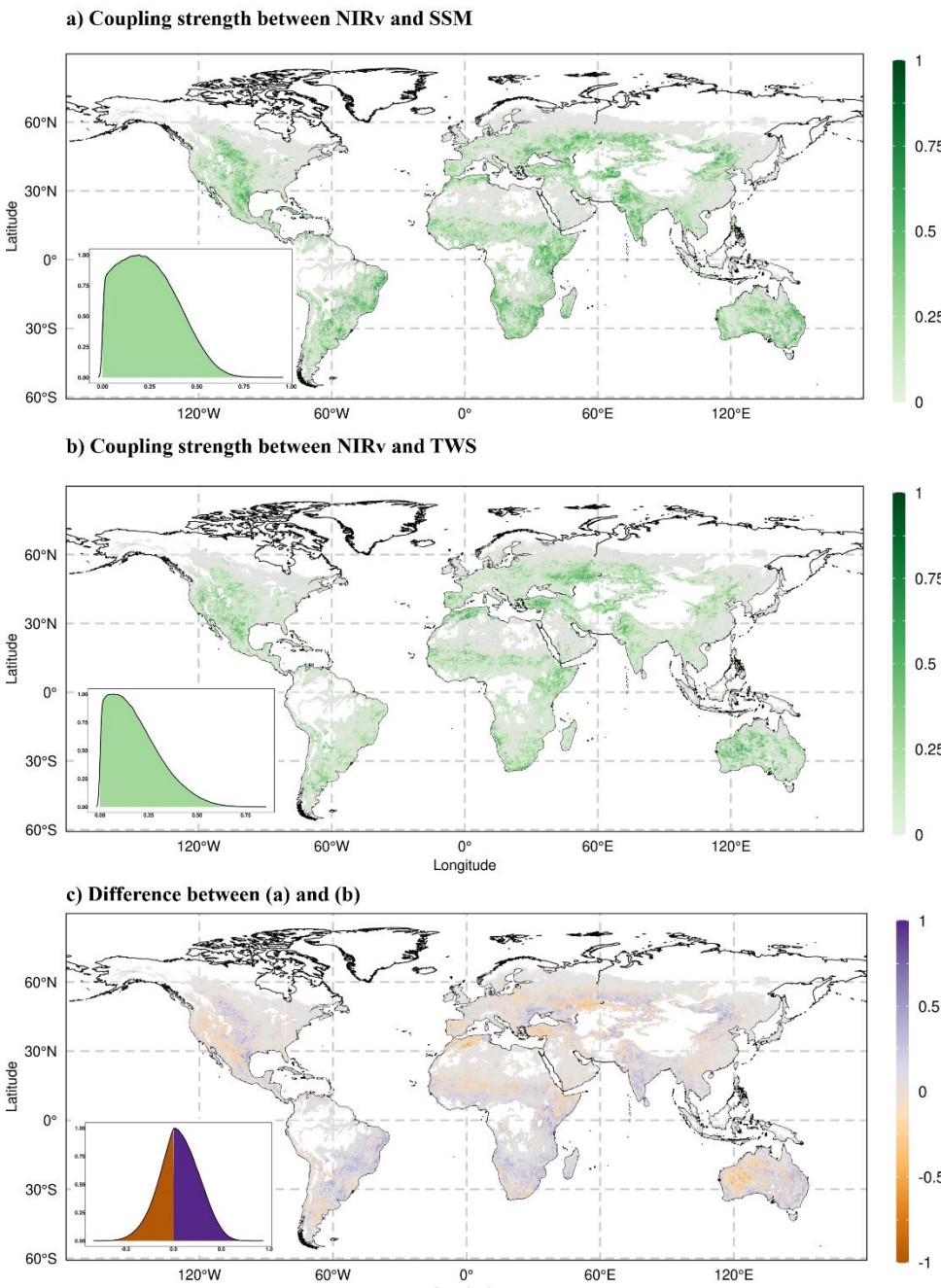

**Figure 1: Coupling strength between vegetation functioning (NIRv) and (a) near-surface soil moisture (SSM), and (b) total water**
**storage (TWS) during the growing season months. The color bar denotes the mean partial correlation for each grid cells, computed**

from the partial correlations across individual bootstrapping samples. Monthly anomalies of all variables are used to calculate the partial correlation. (c) Difference between (a) and (b). The purple colour in (c) indicates the greater partial correlation of NIRv with SSM compared to the partial correlation of NIRv with TWS while orange colour indicates the opposite. Grid cells with positive relationships for both correlations (a) and (b) are shown in (c) with blueish and orange colours. Light grey colour indicates negative partial correlations between NIRv and water storage. The absence of color within the land boundary signifies inadequate observational data for precise computation of the partial correlation. Each inset in the respective maps illustrates the probability distribution function (pdf) of the correlations.

The partial correlation of NIRv with near-surface soil moisture varies globally during growing-season months (**Figure 1a**). NIRv demonstrates stronger correlation with near-surface soil moisture within semi-arid climates, Central North America, South America, regions in South Africa and Australia. The correlation is stronger in Southern Europe and the Mediterranean region compared to central and Northern Europe. The correlation gradient from the hot and dry Mediterranean region to wet and cold Northern Europe corresponds to the gradient of water-limited ecosystems to energy-limited ecosystems obtained in other studies (Denissen et al., 2022; Teuling et al., 2009).

The global correlation of NIRv with TWS follows a similar pattern as with SSM (**Figure 1b**) in growing-season months. The correlation of NIRv with TWS is higher in drier central northern America and Australia compared to other regions. The similarities in the correlation of NIRv with SSM and TWS are expected because the monthly anomalies of SSM and TWS are highly correlated during growing season months in most of our study area **(Figure S2).**

The difference between the partial correlation of NIRv with SSM and TWS (**Figure 1c**) indicates that the NIRv correlates stronger with TWS in Western America, Southern Europe, and arid regions of Australia compared to other regions globally during growing-season months. In South America and Southern Africa, however, the NIRv shows a stronger correlation with SSM. To ensure that the observed patterns of difference of partial correlation between SSM and TWS are not the artifacts arising from the computation of differences based on mean partial correlation, we compared the 95% confidence intervals obtained through bootstrapping. Our results indicate that, for the majority of the considered grid cells, the entire confidence intervals of the correlation (NIRv ~ TWS) fall outside the bounds of the correlation (NIRv ~ SSM) which indicates that the correlations differences are significant, thus enhancing the robustness and confidence in our findings (**Figure S4**). Furthermore, even if we control for the effect of soil water storage (SSM or TWS) when computing partial correlation to discern the relative importance for vegetation, it should be noted that the varying noise levels inherent in these datasets might impact our results.

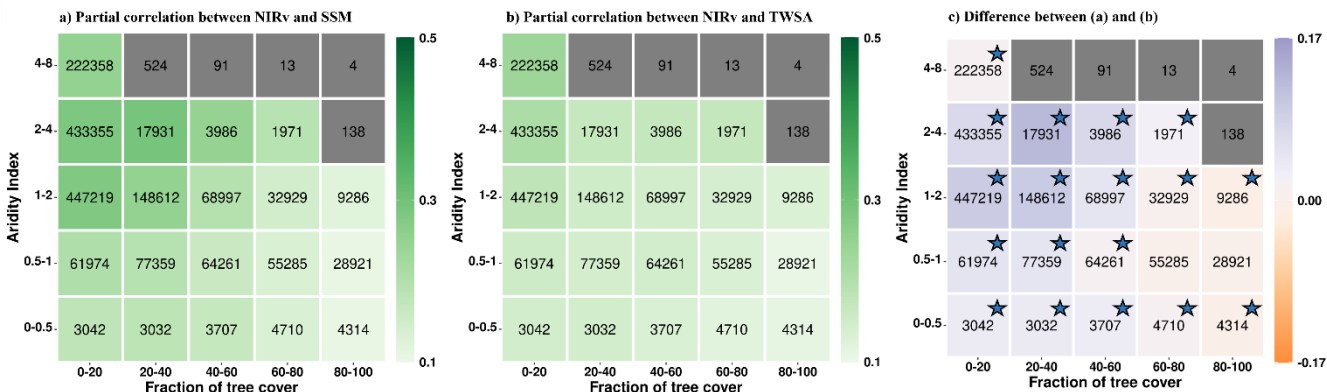

Figure 2: Summarising the coupling strengths of vegetation functioning (NIRv) with (a) near-surface soil moisture (SSM) and (b) terrestrial water storage (TWS) in the growing season-months across climate (aridity index) and vegetation regimes (fraction of tree cover). (c) shows the difference between (a) and (b). Numbers within the boxes denote the number of grid cells for each aridity-tree cover class. Aridity-tree cover classes containing less than 1000 grid cells are shown in grey. The color bar denotes the mean partial correlation for each class, computed from bootstrapping. The asterisk in figure (c) signifies that the 95% confidence interval (lower and upper) shares the consistent sign (+/-) in the difference of partial correlation. Only grid cells with positive partial correlation are considered.

Next, we analyse the partial correlation between NIRv and soil water storages across different aridity and tree cover fraction classes during growing season months. For this, we group the grid cells into different aridity and tree cover fraction classes and then do bootstrapping to compute mean partial correlation and the 95 percent confidence intervals for each class with more than 1000 grid cells. We find that the partial correlation of NIRv with SSM (**Figure 2a**) increases with increasing aridity for aridity index (0-4). This can be attributed to the intensification of water stress on vegetation under increasingly arid conditions, resulting in a stronger correlation between NIRv and SSM. However, for a further increase in aridity (4-8), the strength of the correlation of NIRv with SSM declines. This is due to a low soil moisture availability and low temporal variability under extremely arid conditions (**Figure S5**). The pattern of increasing correlation along aridity index is also observed in the partial correlation between NIRv and TWS (**Figure 2b**).

Furthermore, the correlation of NIRv with SSM decreases for higher tree cover fractions **(Figure 2a)**. However, such a gradient along tree cover fraction is less pronounced in the partial correlation of the NIRv with TWS (**Figure 2b**). This overall depicts that the coupling of vegetation functioning with SSM is generally higher for non-forested areas compared to forested areas while this gradient is less pronounced in the case of TWS.

Though the difference in inherent noise levels associated with SSM and TWS impacts partial correlation analysis, we can compare the evolution of the gradient along tree cover or aridity index and assert how the relevance of SSM and TWS changes with varying tree cover or aridity index, assuming that the noise levels are similar across varying AI-TC classes. Taking this into account, we find that NIRv correlates more strongly with near-surface soil moisture compared to terrestrial water storage in semi-arid regions with low tree cover (**Figure 2c**), suggesting that the vegetation preferentially takes up water from SSM

whenever available to meet its transpiration demand. This might be due to lower energy expenditure on root water uptake, abundant nutrients and reduced chance of root water logging in the near-surface soil moisture (Feldman et al., 2023; Schenk and Jackson, 2002; Tao et al., 2021). Conversely, the correlation between the NIRv and TWS in arid areas (AI 4-8) and regions with a high fraction of tree cover is equivalent to or greater than that of SSM, suggesting that trees can utilise their extensive root systems to access deeper soil moisture, as also observed in arid vegetation. This is consistent with previous studies reporting that the vegetation dependence on sub-surface soil moisture is higher in arid and seasonal-arid climates (Miguez-Macho and Fan, 2021). However, in certain regions with higher tree cover in humid areas, specifically with AI 0.5-1, such conclusions cannot be confidently drawn statistically. The reason is that the confidence intervals for the difference in partial correlation of NIRv with SSM and TWS fluctuate between positive (indicating greater relevance of SSM) and negative (indicating greater relevance of TWS) values (**Figure 2c**).

Note that while our analysis focuses on regions with water-controlled vegetation as denoted by positive correlations between NIRv and the considered soil water storages, some of these grid cells are located in comparatively wet climate regimes with aridity index values between 0 and 1 (**Figure 2**). This highlights the relevance of non-climatic factors such as soil and vegetation types or topography in determining vegetation-water relationships in addition to the climate regime. Next to this, in **Figure 2c** it seems that the relevance of terrestrial water storage is comparatively higher in wet climate (aridity 0.5-1) than in transitional climate regimes (aridity 1-2) as shown with the smaller correlation differences. This, however, is probably not the case and might simply be a reflection of reduced variability in surface soil moisture (**Figure S5**).

### 3.2 Coupling of vegetation functioning with surface soil moisture and total water storage in dry months

The correlation between NIRv and soil water storage increases during dry months (**Figure 3a,b**) compared to growing season months (**Figure 2a,b**). This increase is consistent for both SSM and TWS and across all tree cover fractions and aridity classes. This is because the water limitation on vegetation increases in dry months and so does the vegetation's sensitivity to the moisture. During the dry months, the correlation with near-surface soil moisture tends to rise, but the correlation with terrestrial water storage increases even more significantly **(Figure 3c)**. This indicates the relevance of deeper water resources during periods of scarce rainfall. The partial correlation maps **(Figure S6)** also reveal that NIRv's correlation with TWS increases more than its correlation with SSM for most grid cells.

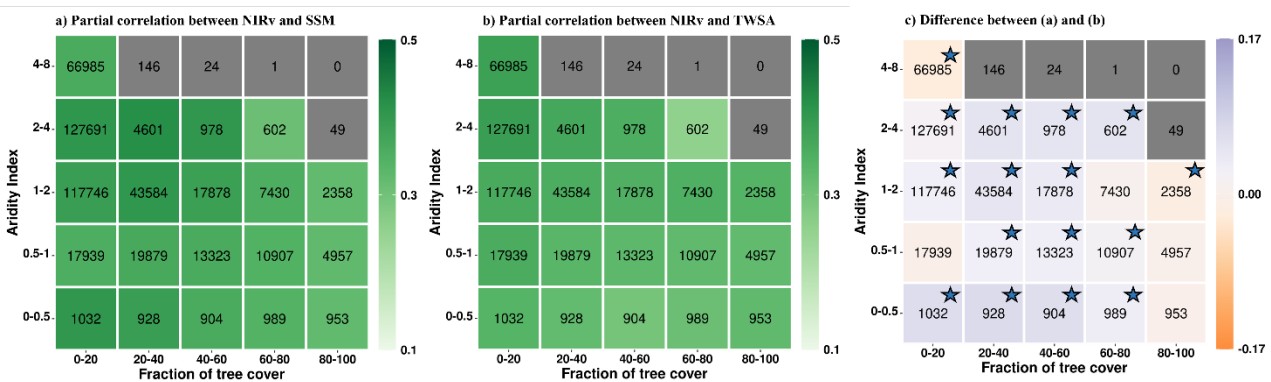

**Figure 3: Summarising the coupling strengths of vegetation functioning (NIRv) with (a) near-surface soil moisture (SSM) and (b)**
**terrestrial water storage (TWS) in the 10% driest months in each grid-cell across climate (aridity index) and vegetation regimes**
**(fraction of tree cover). (c) shows the difference between (a) and (b). Numbers within the boxes denote the number of grid cells for**
**each aridity-tree cover class. Aridity-tree cover classes containing less than 1000 grid cells are shown in grey. The color bar denotes**
**the mean partial correlation for each class, computed from bootstrapping. The asterisk in figure (c) signifies that the 95% confidence**
**interval (lower and upper) shares the consistent sign (+/-) in the difference of partial correlation. Only grid cells with positive partial**
**correlation are considered.**

During dry months, the number of analysed grid cells (**Figure 3**) is lower compared to all growing season months (**Figure 2**).

We performed a reanalysis of the correlation patterns within aridity-tree cover classes by selecting only those grid cells that

displayed positive partial correlation between NIRv and soil water storages during both the dry months and the growing season

339 months. The results demonstrate that the previously observed patterns remain valid, thereby eliminating the impact of the

340 differing numbers of grid cells analysed (**Figure S7**).

**3.3 Climate, vegetation, and topographic controls on the relevance of surface soil moisture vs. total water storage on vegetation**

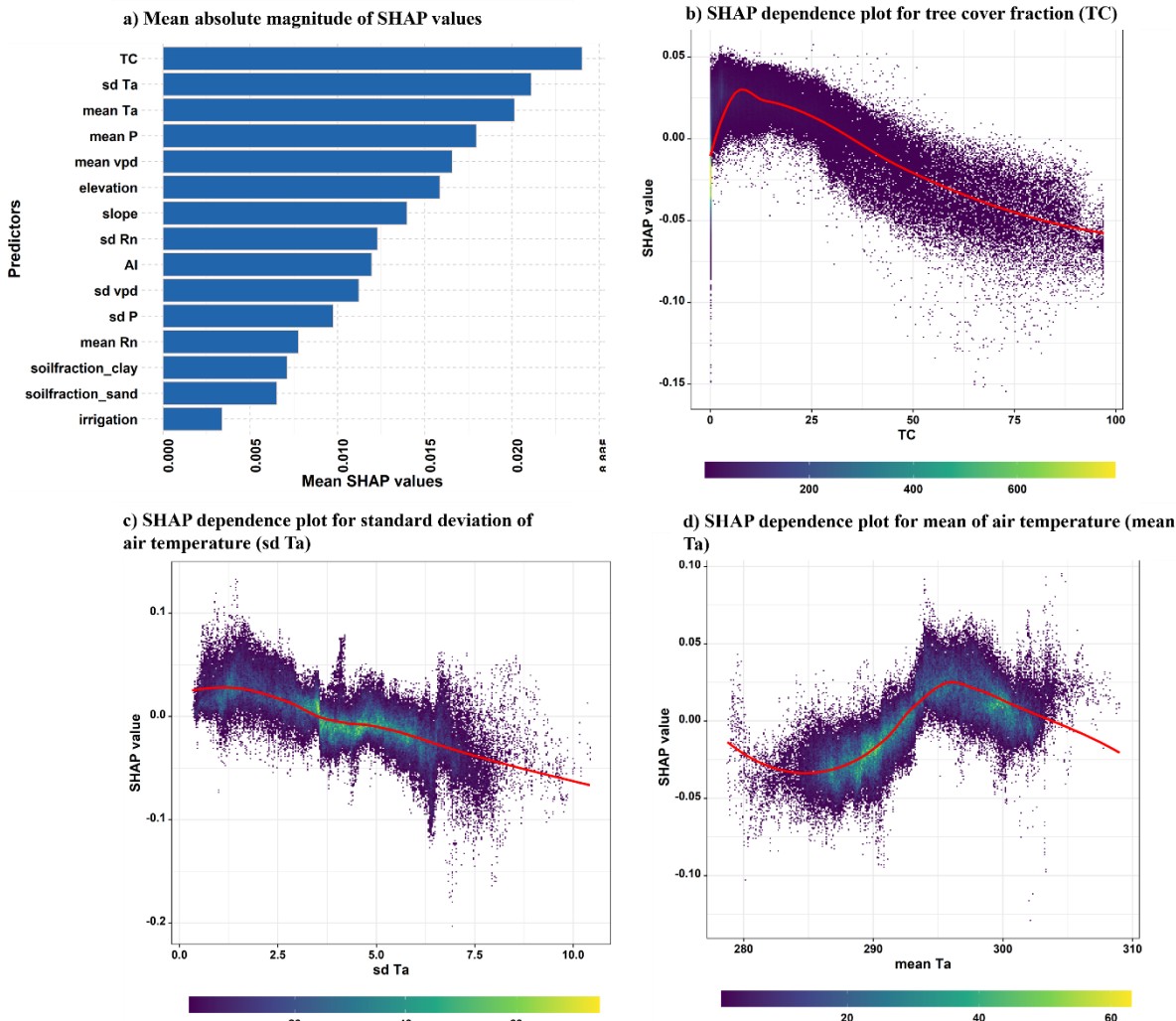

**Figure 4: (a) Global feature importance based on the mean absolute magnitude of the SHAP values. The higher the mean SHAP values, the greater the predictor's relevance. (b-d) Evaluation of SHAP values (=contributions to the correlation difference illustrated in Figure 1c) against predictor values for the 3 most relevant predictors tree cover fraction (TC), variability of temperature (sd $T_a$) and mean temperature (mean $T_a$) during the growing season months. The colour indicates the density of data points. For plotting (b), (c) and (d), only 10 percent random samples of the whole dataset are utilised.**

We use a random forest model to understand the spatial variability in the relevance of SSM versus TWS for NIRv. The model was trained with 15 climatic, vegetation, and topographic predictors against the target variable which is the difference of the partial correlations of NIRv with SSM and TWS during growing season-months ($R^2 = 0.59$, see **methods section 2.2.3**). The mean absolute SHAP value plot shows that the tree cover and the climate variables (mean and standard deviation of $T_a$) are most important variables for explaining the spatial variability in the relative importance of SSM vs. TWS for vegetation

functioning (**Figure 4a**). This overall highlights that the relative importance of SSM vs. TWS for the vegetation is broadly controlled by vegetation type, reflecting the local adaptation of ecosystem and climate, influencing water availability (Stocker et al., 2023).

Tree cover fraction is an important factor in determining the relevance of SSM and TWS for vegetation functioning (**Figure 4b**). Regions with a high tree cover are more dependent on TWS, as trees generally have deeper root systems that allow them to adjust water uptake between different depths (Tao et al., 2021). Grasslands on the other hand have shallow roots that are more susceptible to surface soil moisture variations (Yang et al., 2014).

Similarly, the relative importance of SSM and TWS varies non-linearly with the mean growing season temperature (**Figure 4d**). TWS tends to be more crucial for vegetation functioning in areas with low (approximately below 20°C) or high (above 27°C) growing season temperatures, while SSM has greater importance in regions with moderate growing season air temperatures. One possible explanation for this trend is that high temperatures induce a strong atmospheric water demand that dries near-surface soil layers, which leads vegetation to increase water extraction from deep soils. This observation is further underscored by the analogous pattern observed in the SHAP dependence plot for vpd, which accentuates atmospheric water demand (**Figure S8b**). In contrast, SSM is more available during growing season months in regions characterised by moderate temperatures. We hypothesize that the regions that experience relatively cold growing season temperatures exhibit stronger temperature and weather variability that may contribute to longer dry periods and, thus, emphasises the importance of deeper soil moisture for vegetation functioning. However, it should be noted that our findings regarding the relevance of TWS at high temperatures must be interpreted with caution due to the exclusion of most tropical forest regions from our analysis (**Figure S9**). As a result, most warm regions are dry, and there are only a few hot and wet regions included in our training data.

Not only the mean of the growing season temperature, but also its variability is crucial for explaining the significance of SSM and TWS for vegetation functioning (**Figure 4c**). A higher temporal variability in temperature increases the importance of TWS for vegetation. This is because atmospheric water demand scales with temperature. Hence, higher variability in temperature implies more peaks in related atmospheric water demand which is a stronger incentive for plants to access deeper water storages which are more often available to meet the vegetation´s transpiration demand.

**Figure S8** illustrates the effect of the other six important predictors on the model output. Apart from climatological parameters (mean P, mean vpd, variability in $R_n$, and aridity index ), elevation and slope explain part of the variability in the relevance of SSM vs. TWS for NIRv. Although the reasons for increasing relevance of TWS for vegetation functioning at higher elevation remain unclear, it may be due to elevation's strong correlation with other climatic variables such as $T_a$ and P.

Several local studies identified other relevant factors that determine root water uptake depth such as forest stand age and tree height, competition, root hydraulic architecture, and tree species (Zhu et al., 2022; Quijano et al., 2012; Stahl et al., 2013, Gessler et al., 2021; Liu et al., 2021). For example, young trees more easily increase their root activity in the shallow or deep soil dependent on soil moisture than mature trees (Zhu et al., 2022; Drake et al., 2011). These variables were not included in our attribution analysis, because they are not available at global scale.

**3.4 Robustness Tests**

In the aforementioned analysis, we included grid cells exhibiting both positive partial correlations, whether significant or non-significant. Upon further examination, we specifically assessed the evolution of partial correlation between NIRv and water storages, considering only grid cells with significant partial correlation ($p < 0.05$). The observed patterns along the aridity-tree cover gradient remained similar during growing season months. This suggests the robustness of our results to the choice of the statistical significance criterion, albeit with a substantial reduction in the number of globally available grid cells when considering only significant partial correlation (**Figure S10**).

Furthermore, to ensure that our results are robust to variations in the threshold for Solar-Induced Fluorescence (SIF) used to define growing season months, we conducted additional analyses with a different SIF threshold. Instead of filtering out all months from 2007-2018 when the mean-monthly SIF value was below the threshold of 0.2 mW/m²/sr/nm, we utilized a threshold of 0.5 mW/m²/sr/nm. Elevating the SIF threshold implies the exclusion of additional months characterized by lower vegetation activity for the partial correlation analysis. However, it is essential to note that this threshold does not seem to affect the number of globally available grid cells during growing season months and hence patterns along AI-TC classes are similar. Instead, it specifically influences the selection of dry months and hence the number of grid cells available for the analysis during dry months. Nevertheless, even with the elevated SIF threshold for defining growing season months, the observed patterns along aridity-tree cover (AI-TC) classes remain largely consistent with the results obtained in our main analyses (**Figure S11**).

Although NIRv can largely reflect vegetation functioning (Badgley et al., 2017), we repeat our analysis with SIF, which is an alternative and independent indicator for vegetation functioning and shows a near-linear relationship with gross primary productivity at the ecosystem level (Guanter et al., 2012). However, SIF is only available at a coarse resolution of 0.5 degree. The partial correlations, r(SIF~SSM) and r(SIF~TWS) largely agree with the pattern of r(NIRv~SSM) and r(NIRv~TWS) across varying aridity index and tree cover classes (**Figure S12**). This suggests that our overall conclusion on the relevance of SSM or TWS for vegetation functioning is robust across different indicators of vegetation productivity.

Additionally, we tested if our results are robust when the aridity index is calculated based on the FAO Penman-Monteith Reference Evapotranspiration equation, for which we applied aridity classification based on UNEP 1997 guidelines. Our results

confirm the findings of **Section 3.1** and **Figure 2** that as aridity increases, the correlation of NIRv with near-surface soil moisture (SSM) and total water storage (TWS) intensifies. Moreover, in hyper arid regions (AI < 0.03) the correlation with TWS surpasses that with SSM (**Figure S13**). They also confirm that regions with higher tree Cover (TC) fraction correlates more strongly with TWS compared to SSM. Thus, the choice of aridity index formulation does not alter our main conclusions.

When analyzing partial correlations between Total Water Storage (TWS) and vegetation metrics (NIRv or SIF) at finer resolutions (0.05 degrees for NIRv or 0.5 degrees for SIF), it is crucial to acknowledge the potential emergence of significant spatial autocorrelation. This is attributed to the fact that the actual spatial resolution of the satellite signal underlying the TWS data is 2-3 degrees.

**4. Summary and Conclusions**

In this study we compare the relevance of near-surface soil moisture and of terrestrial water storage for vegetation functioning across the globe. We find that in semi-arid regions and regions with low tree cover, vegetation preferentially utilises the water from shallow soil, which is related to continuous availability of near-surface water availability and lack of deep rooting systems respectively. The stronger correlation of NIRv with SSM than TWS is supported by site-level studies that find a higher root water uptake of surface soil moisture (Brinkmann et al., 2019, Gessler et al., 2021, Deseano Diaz et al., 2023; Kulmatiski and Beard, 2013), also when deeper water is available. Some local studies however find a higher root water uptake from deeper layers (Zhu et al., 2022).

By contrast, in mostly forested regions and in relatively dry climate regimes, the correlation with terrestrial water storage is comparable or higher than with near-surface soil moisture, indicating that trees and vegetation in arid regions use their deep root systems to access deeper soil moisture. Point-scale studies also found a different water uptake depth for trees and grasses for example in savanna ecosystems (Kulmatiski et al., 2010), and a different water uptake depth for tree species (Kahmen et al., 2022). Liu et al. (2021) showed for example that for a karst forest in Southwest China, evergreen species rely mostly on water sources from the 0-30 cm layer, while deciduous species extracted most water from the 30-70 cm layer.

We also find that vegetation's preferential water uptake depth changes over time. During particularly dry months, the relative importance of terrestrial water storage is higher, highlighting the importance of deep water resources during periods of low soil water availability. This is in line with previous studies showing changes in vegetation's water uptake depth during drought periods at small spatial scales where accessing water in deeper soil layers helps plants to alleviate water stress and maintain transpiration (Migliavacca et al., 2009; Tao et al., 2021).

Our global results are supported by site-scale studies that find that, during drought, the deeper roots play a more active role in water extraction (Stahl et al., 2013, Volkmann et al., 2016; Tao et al., 2021). In some studies however, the increase of deep water uptake is only relative: the absolute uptake of deep water does not increase, but the uptake of shallow water decreases (Brinkmann et al., 2019, Gessler et al., 2021, Rasmussen et al., 2020; Kühnhammer et al., 2023). This means that the uptake of deeper soil layers cannot compensate for the loss of water uptake from the dry topsoil. Contrary to trees, grasses do not shift their uptake depth (Deseano Diaz et al., 2023), or even extract water from the most shallow soils (Prechsl et al., 2015, Kulmatiski and Beard, 2013).

Furthermore, we show that the spatial variability of the importance of near-surface soil moisture vs. terrestrial water storage for vegetation functioning is influenced by fraction of tree cover and mean and standard deviation of air temperature. This emphasises the role of climate in determining shallow vs. deep soil water resources, and the role of vegetation in adapting to different soil water availability patterns.

Vegetation functioning and soil water storages are generally coupled in both directions, i.e. while soil moisture availability affects vegetation functioning (positive coupling), this in turn also affects soil moisture through transpiration (negative coupling). As our study focuses on water-controlled vegetation we only consider positive couplings and filter out grid cells with negative correlations. Future research may consider the relevance of soil moisture across depths for the negative coupling regions.

Overall, our analysis illustrates that satellite-based data can be used for belowground analysis at large spatial scales thanks to the fact that satellite retrievals can assess soil water storage dynamics across depths and because vegetation in water-controlled areas can be used as an indicator of soil water dynamics. Such novel ways to improve our understanding of belowground water dynamics is necessary and valuable as respective in-situ observations are scarce and of limited representativeness for larger areas, particularly given the typical spatial heterogeneity of soils and vegetation. Our results can further inform a better representation of belowground processes in global models in order to support more accurate projections of future changes in climate, water resources, and ecosystem services.

**Data availability**

The monthly SIF data is available from https://www.gfz-potsdam.de/sektion/fernerkundungund-geoinformatik/projekte/global-monitoring-of-vegetation-fluorescence-globfluo/daten.The NIRv was calculated from the red and near-infrared reflectance obtained from the MOD13C1 v006 product (https://lpdaac.usgs.gov/products/mod13c1v061/). The ESA-CCI soil moisture can be accessed through https://esa-soilmoisture-cci.org/ and Terrestrial Water Storage Anomaly data can be accessed through https://podaac.jpl.nasa.gov/dataset/TELLUS_GRACGRFO_MASCON_CRI_GRID_RL06_V2.

The ERA5 climate variables are available from https://www.ecmwf.int/en/forecasts/datasets/reanalysis-datasets/era5 . Tree

cover fraction data is available from the AVHRR vegetation continuous fields products

https://lpdaac.usgs.gov/products/vcf5kyrv001/, land cover data is available from https://www.esa-landcover-cci.org/, and

topographic data is available via https://www.earthenv.org/topography. Similarly, the irrigation fraction data could be accessed

from https://mygeohub.org/publications/8 .

**Competing Interests**

The contact author has declared that none of the authors has any competing interests.

**Acknowledgements**

The authors thank Ulrich Weber for help with obtaining and processing the data, Sujan Koirala for valuable scientific and

technical support and the Hydrology–Biosphere–Climate Interactions group at the Max Planck Institute for Biogeochemistry

for fruitful discussions. Prajwal Khanal, Anne Hoek van Dijke and Rene Orth acknowledge funding by the German Research

Foundation (Emmy Noether grant no. 391059971).

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
