# Peer review of "Relevance of near-surface soil moisture vs. terrestrial water storage 1"

_EGUsphere, 2023_

## Author Comment (AC2)

[Figure]

**Figure R1: Summarising the coupling strengths of vegetation functioning (NIRv) with (a) near-surface soil moisture (SSM) and (b) terrestrial water storage (TWS) in the growing season-months across climate (aridity index) and vegetation regimes (fraction of tree cover). The only difference with Figure 3 is the different formulation of AI used in this figure.**

**Figure R2: Global feature importance based on the mean absolute magnitude of the SHAP values (including vpd).**

[Figure]

**Figure R3: Summarizing the ratio of r(NIRv~TWS_dry) - r( NIRv~TWS_growing) to r(NIRv~SSM_dry) - r(NIRv~SSM_growing) across aridity-tree cover classes.**

---

## Author Response (AR1)

**Khanal et al evaluate the relative controls of surface and deeper soil moisture on vegetation using mostly satellite data from CCI soil moisture and GRACE terrestrial water storage. They argue that surface soil moisture controls vegetation generally more than deeper moisture stores depending on climate, but deeper moisture stores increase their control in drier months. I have a positive outlook on the study. It evaluates an unexplored gap about the depth of vegetation water use which remains a highly uncertain part of the biosphere. The use of satellite retrievals was highly appropriate for the (nicely posed) objectives. I do think the study needs more work. I have several concerns about the statistical analysis and interpretation of results that should be addressed. I also think more context needs to be added about the uncertainty of the depth of representation of GRACE and CCI and how their correlation confounds interpretation of results. Because of these issues, I think some of the current conclusion arguments extend beyond what we can say from the analysis and may need to be tempered somewhat. Nonetheless, even with the tempered arguments, the study is very insightful and can be a great contribution to the community. See comments below.**

**-Andrew Feldman**

We thank the reviewer for his positive feedback on our manuscript and thank him for the constructive comments to improve our manuscript.

**Major Comments**

1. **Is it possible to determine whether CCI or GRACE controls vegetation more only from partial correlations? In reading L246-257, I am wondering if one can conclude greater control from correlation differences that are within 0.1 of each other, especially when the partial correlations do not control for the other soil moisture metric. Much mutual information is thus involved which is difficult to disentangle. Is there some significance metric that can be used to denote the difference in correlations? Or another different metric entirely? At what point do we say that both depths are controlling vegetation in an indistinguishable manner? Ultimately, I think some main conclusions that one depth controls vegetation more than another are not fully supported by the correlations. At least we can say that these variables are largely interconnected and both surface and TWS moisture have some control on vegetation in many places, which is an interesting finding in itself!**

We appreciate the reviewer's insightful query regarding the possibility of ascertaining whether CCI or GRACE exerts a stronger influence on vegetation, particularly when relying solely on partial correlation. This is especially pertinent when we do not control for the other soil moisture metric, as this introduces mutual information, which can be challenging to disentangle. In order to bolster the robustness of our findings, we performed two distinct analyses.

In the first analysis, we computed partial correlations between NIRv and surface soil moisture (SSM) (as well as NIRv and terrestrial water storage (TWS)), while controlling for TWS (or SSM) alongside energy variables (Ta and Rn). This approach enables us to examine the correlation of NIRv with each

water variable independently, while accounting for the influence of other water variables. However, it's important to note that introducing additional control variables, which might confound the correlation, results in a reduction in the number of grid cells displaying significant partial correlations between NIRv and water variables, and it also diminishes the strength of both partial correlations. With the new approach, the number of grid cells available was too little to analyze the correlation difference and make inference at the global scale. Therefore, we omitted the significance criteria and focused on all grid cells demonstrating a positive correlation during the growing season months for this particular analysis. We will replace Figure 1 with the new partial correlation maps that take these additional control variables into account and move the former Figure 1 to Supplementary material.

Even with these adjustments, the global spatial patterns in the partial correlation maps, specifically r(NIR~SSM/T,R and TWS), as well as r(NIRvTWS/T,R and SSM), remain largely consistent with those obtained without considering the additional control variables, which is r(NIRv~SSM/T,R) and r(NIRv~TWS/T,R) respectively. This underscores the robustness and confidence in the global spatial patterns of correlations between NIRv and water variables, even after untangling the effects of other water variables.

In the second analysis we conducted a bootstrapping analysis. This analysis allowed us to compute bootstrap means and 97.5% confidence intervals for the difference in partial correlation between r(NIRv~~SSM) and r(NIRv~TWS) for each AI-TC class. We will add the outcome of the bootstrapping analysis to Figure 2.

Our second analysis reveals that, throughout the growing season months, the bootstrapping results show that for most random samples drawn from each AI-TC class, the sign of the correlation difference is the same as for the overall analysis, underlining the robustness of our findings. This consistent positive result underscores a stronger correlation between NIRv and SSM compared to TWS during the growing season months, even after accounting for the influence of other water variables.

Conversely, during the dry season months, the majority of AI-TC classes displayed positive confidence intervals. This observation highlights the greater relevance of SSM for NIRv compared to TWS during the dry season months within these classes. However, the bootstrapping analysis also confirmed the higher relevance of TWS in specific classes. This pattern implies a stronger correlation of NIRv with TWS compared to SSM.

It's worth noting that in certain classes, the confidence intervals did not provide a conclusive distinction, indicating that the TWS and SSM are similarly relevant for the vegetation in these regions in dry months.

Apart from this, we agree with the reviewer that both surface and deeper moisture have some control on plant growth, and we will stress this more in the manuscript.

Lastly, to address the reviewer's suggestion that the primary conclusion, which posits that one depth level exerts greater control over vegetation compared to another, lacks full support from correlations, we will modify the sentence in the abstract as follows: "We find that vegetation functioning seems generally more strongly related to near-surface soil moisture, particularly in semi-arid regions and

areas with low tree cover. At the same time we note that this comparison is hampered by different noise levels in these satellite data streams."

**2. Discussion early on in the manuscript is needed about depths of representation for both ESA CCI soil moisture and GRACE TWS. There is still much debate about what layers these products represent as well as what layers vegetation uses water from. I would provide context about both of these points in the introduction. (Trying to avoid self promotion but please see references within this WRR study, already referenced in preprint form in this paper, that can help with supporting this discussion: https://agupubs.onlinelibrary.wiley.com/doi/full/10.1029/2022WR033814)**

Thank you for sharing the paper, which is highly relevant to the study of soil moisture and vegetation uptake depth. The paper emphasizes that satellite-driven soil moisture data obtained through microwave remote sensing provides information about soil moisture at greater depths than traditionally considered, both in wet and dry conditions and hence capture water content relevant for studying vegetation water uptake. We will reference this study when discussing vegetation uptake depth in paragraph 2 of the introduction and when explaining our results. Regarding GRACE TWS, it represents the total water content, making it challenging to determine specific depths. As it has been found to exhibit a strong correlation with vegetation activity, we believe it still offers an implicit reference to water content relevant for vegetation, extending beyond surface soil moisture as captured by ESA CCI. However, under specific conditions explained in Line 77.

**3. Following the previous point, CCI soil moisture (mostly from C and X band? Please clarify) is theoretically only providing microwave emission from the top 2 cm. Most would argue that plants are sensitive to moisture at much deeper depths. What does is mean that CCI soil moisture is correlated to vegetation behavior? This link needs to be made early on for the reader. For example, surface soil moisture is often correlated with rootzone soil moisture layers below effectively providing more information than only what is shown in 0-2cm layer. Therefore, this correlation can mean 0-2cm moisture are contributing to root water uptake and/or deeper layers correlated to 0-2cm moisture are contributing to root water uptake.**

The reviewer mentions a valid point which we will clarify in the manuscript. Even when vegetation activity is well correlated with surface soil moisture, it is challenging to precisely define the depths relevant for vegetation uptake as surface soil moisture often shows a strong correlation with rootzone soil moisture, especially during wet months.

However, our aim is to investigate how vegetation uptake depths vary across different vegetation types and from growing season months to dry periods on a global scale, rather than identifying which exact soil depth is most related to vegetation activity. And while the soil depth of our two dataset (CCI and GRACE) partly overlaps, they do allow for separating surface moisture from overall water availability using partial correlation.

**4. One interpretation that could be made clearer (based on findings such as in L213-216) is that both surface and deeper moisture sources can control vegetation. Additionally, when both correlations with TWS and surface soil moisture are found to be positive, it is also not clear if**

**these are both depths controlling vegetation. Or only one of the depths controls the vegetation but a positive correlation shows up in both because TWS and surface soil moisture are correlated to one another. This point is especially a concern with TWS and surface soil moisture not controlled for in the same regression.**

Our improved methodology, where the partial correlations control for the other variable, holds promise in addressing this challenge and enhancing our understanding of variations in vegetation uptake depths.

**5. I know it is difficult to test, but could the results change if at a daily timescale? Might we expect TWS and surface soil moisture to be more correlated at a monthly scale and therefore more difficult to pull apart which is controlling vegetation at a monthly scale? Some speculation on this and discussion of result dependence on temporal scale is necessary.**

Our study was constrained by the unavailability of daily TWS data. On the other hand, the processes that impact the correlation between NIRv and soil moisture act at longer than daily time scales. For example, the drought effects on NIRv are lagged, and the soil moisture data has a high autocorrelation. Though, performing the analysis at the daily time scale might be beneficial for disentangling TWS and SSM dynamics, a potential drawback would be we could not capture the soil water- vegetation relationship in this scale because of the time lags and soil moisture memory as mentioned before. So, thoughts on changes in correlation strength for the daily time scale would therefore be very speculative, and we prefer to stay away from that in the manuscript.

**6. Should the analysis be conducted with anomalies? The seasonal cycle could be larger in either surface soil moisture or TWS which may contribute to a higher correlation in that variable with vegetation, which is not related to the direct control on vegetation. I know this is partially mitigated by using the growing season but be aware that the seasonal cycle can spuriously inflate the partial correlations.**

For our manuscript, we calculate the correlation between the monthly anomalies of vegetation activity and anomalies in water storage. It is important to note that we have accounted for the removal of the long-term mean monthly cycle and linear trends, as described in section 2.2.1 and illustrated in Figure S1.

Specific comments

**L71-79: It would be helpful to expand on what depths TWS represents from previous literature. I've seen studies noting 1-3m because this is where most of the water variations are that GRACE can detect. This point is uncertain but it would be good to lay out the previous knowledge for the reade**r.

We will mention that GRACE TWS anomalies include variations in all sub-surface water, including the top soil and deep aquifers. We have not yet found any studies that are more explicit about this, but we will do another search for literature when revising the manuscript.

**L83: really great questions**

Thank you!

**L139: Since GRACE is 0.5 degrees, I recommend conducting the analysis at 0.5 degrees. Some errors may otherwise arise in attribution with higher resolution datasets with the SHAP analysis**

We are uncertain about the potential impact of this on our SHAP analysis results. Given that the resolution of the other variables used in the SHAP analysis differs from the 0.05-degree resolution (we employed) and the 0.5-degree (suggested), we opted to omit this comparison. Although there may be some scaling effects in our results, we remain uncertain about the extent of their impact and it would be tremendously valuable to ascertain whether SHAP analysis is robust when conducted at different scales.

However, we did perform additional SHAP analyses, as recommended by another reviewer, which involved including vpd and considering partial correlation with constraints of other water variables, as mentioned in our previous response to question 1. These analyses revealed minimal changes in the ranking of the important variables in explaining the correlation differences. Notably, the SHAP dependence plots retained a similar nature, indicating the robustness of SHAP analysis even when controlling for other water storage in computing partial correlation and introducing vpd. We hope this information proves helpful.

**L146: are the results sensitive to this threshold? I imagine this could greatly reduce time periods of the year for arid regions.**

We currently remove all grid cells with a SIF value below 0.2. An increase in this threshold would remove grid cells that are partly vegetated and it would remove months with on average little active vegetation. With a higher threshold we potentially exclude grasslands that have low SIF values compared to forests (Chen et al., 2022). Furthermore, a higher threshold would remove months with lower vegetation activity, and therefore reduce the number of grid cells for our correlation analyses. We therefore remain with our 0.2 threshold.

**L154-155: It would be helpful to show the regression equation(s) for how this was done. I think this is equivalent to a multiple regression with NIRv as the explained variable and the climate variables as explanatory variables.**

We will provide further details on how Spearman partial correlations are computed in the methodology section.

**L157: Both TWS and surface soil moisture have to be negative correlations or insignificant? What if only one of them shows a positive and significant correlation?**

In our manuscript, we excluded data points if one or both of the partial correlations were negative or insignificant. However, for the new partial correlations computed (accounting for the other water storage variables), we exclude only the negative partial correlations (as discussed in our response to

Question 1). We will clarify in the manuscript that we remove any grid cells that have a negative correlation for SSM and / or TWS.

**L162: More detail is needed for how these dry months are determined. Are these the driest of the growing season months within a pixel?**

The dry months were defined based on the lowest 10% value of each grid cell within a pixel, specifically considering the availability of surface soil moisture. These months indeed correspond to the driest periods within a pixel during the growing season. We will clarify this in the methodology section of the manuscript.

**Figure 1: It would be helpful to show the pdf of the spatial distributions so the values can be more easily viewed.**

Thank you for the suggestion. We have adapted this in Figure 1 accordingly.

**Line 206-211: Can this be shown in a figure?**

Figure 1 represents the spatial variability of the correlation that is explained here. Could you clarify what you would like to see in a figure?

**L216: I think Figure S2 would be valuable to show in the main text. This provides a lot of context especially with regard to my major comments.**

Given our focus on emphasizing the correlation between vegetation activity, SSM, and TWS, as well as examining their spatial variations and temporal changes, especially during dry months, we believe that Figure S2, though highly relevant for comprehending the correlation results, it is important to keep in the supplementary section intact. This decision aligns with our aim to effectively emphasize the primary objectives of this paper, while maintaining a smooth and coherent flow of information through figures. Moreover, with our adjusted methodology, in which we incorporate an additional water variable in the partial correlation, it is possible that the correlation between SSM and TWS may be less problematic. We genuinely value your understanding and support in this matter.

**L291: Can max rooting depth be included in the analysis? The Fan et al 2017 PNAS dataset can be used.**

The existing global maximum rooting depth datasets exhibit significant variability and come with substantial uncertainties (Li et al., 2022). Given these challenges, along with the time constraints we face, we have made the decision not to pursue this avenue of investigation.

**L349-351: I think this argument extends beyond what the correlations say. This conclusion is only based on the small correlation differences. Maybe a more specific significance test is needed here. See my major comments.**

We believe that through the major revisions, as outlined in response to major comments 1, we can effectively emphasize and strengthen the conclusion that vegetation preferentially takes up water from the top soil. We agree that we did not show any analyses about rooting depth and we will rewrite the second part of this argument.

**L351-353: What if surface soil moisture and TWS are correlated and both are correlated with vegetation behavior? This leaves open the possibility that vegetation is only accessing shallow moisture but shallow moisture is also correlated with deep moisture, which confounds the results. I don't deny that vegetation is accessing TWS but the conclusion here may need some more evidence based on the current study setup. See my major comment.**

We believe that we have addressed this point with the update of the methodology to take into account the water variables in the partial correlations. As mentioned earlier in our reply, we will discuss and show the strong correlation between SSM and TWS.

**L355-359: awesome finding. Well done.**

Thank you for your appreciation.

**L399: This study is now published in WRR. Please reference that instead of the preprint. https://agupubs.onlinelibrary.wiley.com/doi/full/10.1029/2022WR033814**

We will update this here.

**Citations**

Chen, J., Liu, X., Ma, Y., Liu, L. (2022). Effects of Low Temperature on the Relationship between Solar-Induced Chlorophyll Fluorescence and Gross Primary Productivity across Different Plant Function Types. *Remote Sens. 14*, 3716. https://doi.org/10.3390/rs14153716

Li, W., Migliavacca, M., Forkel, M., Walther, S., Reichstein, M., & Orth, R. (2021). Revisiting Global Vegetation Controls Using Multi-Layer Soil Moisture. *Geophysical Research Letters, 48*(11). https://doi.org/10.1029/2021GL092856

**Reviewer 2**
*Broad Comments*

**This paper evaluates the relative reliance of vegetation on surface soil moisture versus deeper water stores globally using remotely sensed surface soil moisture from ESA-CCI and total water storage from GRACE. This is a very important question with substantial implications for model representations of water movement through the soil-vegetation-atmosphere continuum, and the manuscript provides a novel approach to addressing water uptake depth at the global scale. They find that vegetation functioning (proxied by NIRv) is more strongly related to near surface soil moisture than total water storage anomalies on average, but that deeper water stores are more important in areas with high tree cover and high aridity. These results are robust to using SIF as the indicator of vegetation function, and temporal trends in water uptake depth are comparable to spatial gradients (i.e., TWS becomes more important in arid months).**

**I believe that this paper will make a valuable contribution to Biogeosciences but I have several major concerns regarding the analysis that should be addressed prior to publication:**

We would like to thank the reviewer for the positive feedback and suggestions to improve the manuscript.

1. **The authors calculate the ratio of net radiation to precipitation and use it as an aridity index. This index is used throughout the manuscript to describe trends in the relative strength of the partial correlations of vegetation to SSM vs. TWS anomalies. However, this is a very nonstandard way to calculate aridity index. The authors should: 1) provide a thorough justification for the use of this formulation instead of a standard P/PET aridity index, and 2) demonstrate that the results are consistent if P/PET is used.**

We calculate the Aridity Index by taking the ratio of the long-term mean net radiation (Rn) to precipitation (P), both expressed in millimeters (mm) for each grid cell. We chose this formulation because it provides a direct estimation of aridity and water (energy) limitations on vegetation, eliminating the need to concern ourselves with various formulations used for calculating potential evapotranspiration.

Implementing the reviewer's suggestion, we conducted our analysis using the available Global Aridity Index dataset, which we obtained from **Zomer et al. (2022)**. The dataset can be accessed at the following
https://figshare.com/articles/dataset/Global_Aridity_Index_and_Potential_Evapotranspiration_ET0_Climate_Database_v2/7504448/5

This globally available dataset calculated the Aridity Index using the FAO Penman-Monteith Reference Evapotranspiration equation, with data averaged over the period from 1970 to 2000. We applied an aridity classification that includes:

a.      Humid (AI > 0.65)
b.      Dry sub-humid (AI between 0.5 and 0.65)
c.      Semi-Arid (AI between 0.2 and 0.5)
d.      Arid (AI between 0.03 and 0.2)
e.      Hyper-Arid (AI < 0.03)

This classification is based on the UNEP 1997 guidelines, as detailed in the dataset's manual.

Then, we conducted bootstrapping to calculate the means of partial correlations for the following relationships during the growing season months:(a) NIRv with SSM, (b) NIRv with TWSA and(c) The difference between the above correlations.

In response to a suggestion from another reviewer, we introduced a slight modification. This time, when calculating the partial correlations for NIRv with SSM (TWSA), we incorporated control

variables for TWSA (SSM) along with temperature (Ta), and radiation (Rn). This adjustment enhances our analysis by accounting for additional factors that may influence these relationships.

We observed that the patterns along the Aridity Index (AI) and fraction of tree cover (TC) show significant similarities compared to the figures presented in the manuscript (Figure 2) for growing season months. As aridity increases, the correlation of NIRv with both Soil Surface Moisture (SSM) and Total Water Storage (TWS) also increases. Furthermore, as AI continues to rise, the correlation with TWS becomes stronger than the correlation with SSM. Similarly, with an increase in TC, the correlation of NIRv tends to be higher with TWS compared to SSM. It's important to note that these findings do not alter our primary conclusions and we will incorporate these figures in the supplementary section too.

**2.      Figure S2 demonstrates that, in many areas, TWS anomalies and SSM are highly correlated. The implications of this correlation, and the areas of negative correlation, should be further explored in the text. Disentangling two highly correlated drivers is quite difficult. The limitations of partial correlation should be clearly stated, and some of the associated conclusions should probably be tempered. The time series nature of the data introduces concerns about pseudoreplication as well, which could potentially be addressed through explicit inclusion of the temporal autocorrelation structure in the regressions.**

We thank the reviewer for raising this issue. Since, TWS and SSM anomalies are highly correlated, we slightly updated our additional analysis. We computed partial correlations between NIRv and surface soil moisture (SSM) (as well as NIRv and terrestrial water storage (TWS)), while controlling for TWS (or SSM) alongside energy variables (Ta and Rn). This approach enables us to examine the correlation of NIRv with each water variable independently, while accounting for the influence of other water variables. However, it's important to note that this methodology comes at a cost: introducing additional control variables results in a reduction in the number of grid cells displaying significant partial correlations between NIRv and water variables, and it also diminishes the strength of both partial correlations. With the new approach, the number of grid cells available was too little to analyze the correlation difference and make inference at the global scale. Therefore, we omitted the significance criteria and focused on all grid cells demonstrating a positive correlation during the growing season months for this particular analysis. We will replace Figure 1 with the new partial correlation maps that take these additional control variables into account and move the former Figure 1 to Supplementary material. This will help us to disentangle the effect of highly correlated drivers as raised here.

We will clearly state the limitations of the partial correlation in the methodology section in the updated manuscript.

Regarding temporal autocorrelation, we believe that, on the monthly scale, it may be of less concern for partial correlation between NIRv and water storages compared to the daily or half-monthly scales of analysis. However, we do acknowledge that temporal autocorrelation might be more relevant and impactful at finer temporal resolutions and will highlight these limitations in the discussion section.

**3.      The authors conduct an attribution analysis of the difference in partial correlations of NIRv and SSM versus TWSA using a number of environmental variables, including their aridity index, and mean and standard deviation of temperature, net radiation, and precipitation. They find that mean temperature is the strongest driver of relative reliance of SSM, with relative SSM reliance peaking in areas with moderate temperatures and higher TWS reliance in areas with relatively low or high temperatures. They interpret this temperature effect through the lens of higher atmospheric water demand in very warm regions (lines 306-307), and longer dry periods in cold regions due to weather variability (lines 308-310). First, it**

**is critical to include a direct measure of atmospheric water demand (either VPD or PET) in this analysis, allowing for direct assessment of the mechanism proposed for warm regions. Second, the authors should statistically demonstrate that, in this dataset, cold growing seasons are associated with 1) stronger temperature/weather variability and 2) longer dry periods. The limited importance of the aridity index in this analysis raises additional concerns about the formulation used.**

We thank the reviewer for the interesting suggestion to include atmospheric water demand (vpd) in our SHAP analysis. As suggested, we have updated our analysis with additional predictors, specifically the mean vapor pressure deficit (mean vpd) and the standard deviation of vapor pressure deficit (sd vpd), calculated for the growing season months. These additions will be appropriately reflected in the revised Figure 4 of our manuscript.

Upon integrating vpd into our SHAP analysis, we've observed that the most influential predictors for predicting the difference in partial correlation between NIRv and SSM, and NIRv and TWSA, remain largely consistent as shown in the figure below. The significance of temperature appears to diminish upon the incorporation of VPD, which is logical given their inherent relationship. This observation aligns with our initial assertion that temperature holds physical relevance, as it serves as an indicator of atmospheric water demand. Furthermore, we've found that vpd exerts a non-linear impact on these differences of correlations, which is not shown here but will be updated in the revised manuscript.

**Figure: Global feature importance based on the mean absolute magnitude of the SHAP values (including vpd).**

It's noteworthy that, despite the insightful inclusion of vpd, the performance of our Random Forest (RF) model has experienced a decrease from 0.64 to 0.59 compared to the previous configuration. This might be due to lots of correlated energy drivers like Ta, Rn, vpd, AI on training the model. Nevertheless, the results from SHAP analysis including vpd will be duly reflected in the updated figures and conclusions throughout the manuscript.

*Specific Comments*

**Lines 77-79: Assumptions 2 and 3 should be further justified and referenced. Assumption 2 is particularly concerning in grid cells with highly manipulated hydrologic systems, such as where irrigation results in substantial drawdown of groundwater and reservoir levels throughout the growing season.**

We will update this in the text in the same line with justifications.

**Table 1: Adding the temporal coverage and resolution would be useful (ex. I believe the Seibert data are at 5-year intervals ending in 2005).**

We will revise our table and include the temporal coverage and resolution of our datasets.

**108-115: Can the authors be more specific about the depths represented by the two products?**

We will clarify in the text that 1) CCI SSM is based on microwave measurements that penetrate only the top few centimeters, but that it is representative for a larger depth, and 2) TWS Anomalies include fluctuations in all sub-surface water from the topsoil to deep aquifers.

**Line 115: Citation for the JPL-Mascons product is missing**

We will update this.

**Lines 152 and 165: What percentage of observations were excluded based on the 40-month criteria, and which dataset was the primary limitation here? What percentage of observations**

**were excluded for the dry months analysis based on the 100-month criteria? The justification for using two different sample size criteria for these analyses is not clear from the text. If these sample size cutoffs are somewhat arbitrary, can it be demonstrated that key results are robust to a range of cutoffs?**

We plan to include additional lines in the analysis to explore the percentage of observations that are excluded based on different cutoffs. We will also improve the clarity of our justification for choosing cutoff values of 40 months and 100 months in the text. In addition, we will explicitly detail the methodology section to include information on the percentage of grid cells that were incorporated and the precise method by which this percentage was computed.

**Line 158: Similar to the above comment, what percentage of grid cells exhibited insignificant relationships to soil moisture, and what percentage were negative?**

As in the previous response, we will also explain it in the text in relevant lines.

**Figure 1c legend: Please specify what the white areas represent. Also, the color scale is variously described as blue and purple for positive correlations and red and orange for negative correlations in this legend and the legend for Figure S2; this should be standardized.**

We will specify in the caption that the white areas represent regions with no or insufficient number of data. Apart from this, the references to the figure's colors will be standardized.

**Line 238 and Figure S7: This figure is referenced in text after Figure S2; the SI figures should be reordered to be sequential.**

We will take it into consideration and update our manuscript accordingly.

**Lines 246-247 are redundant with lines 221-222.**

Thank you for catching this. We will remove it in the line 221-222 in the updated manuscript.

**Line 265: Do the authors mean reduced variability in TWS, rather than SSM, as implied by the previous sentence? Differences in the SD of SSM across aridity bins appear to be minimal from Figure S7.**

We agree with the reviewer that the variability appears to be minimal from Figure S7 and we will remove the sentence from the manuscript.

**Lines 275-278: This text indicates that the correlation between TWSA and NIRv increases more between growing season and dry months than the correlation between SSM and NIRv, but the referenced figures (figure 3 and S4) do not show this directly. The ratio of the change in (pcor(NIRv~TWSA)growing season - pcor(NIRv~TWSA)dry months) to (pcor(NIRv~SSM)growing season - pcor(NIRv~SSM)dry months) could be calculated and displayed similarly to the correlation heat maps in Figs 2 and 3, with values greater than one indicating a larger change in the partial correlation for TWSA relative to SSM.**

Figure: Summarizing the ratio of r(NIRv~TWS_dry) - r( NIRv~TWS_growing) to r(NIRv~SSM_dry) - r(NIRv~SSM_growing) across aridity-tree cover classes.

Following the reviewer's suggestions, we computed the ratio of r(NIRv~TWS_dry) - r( NIRv~TWS_growing) to r(NIRv~SSM_dry) - r(NIRv~SSM_growing) in order to provide a clearer representation of the intended information. Values exceeding 1 signify a more pronounced

augmentation in the correlation with Total Water Storage (TWS) in comparison to Soil Surface Moisture (SSM) during the dry season, relative to the growing season.

We find that, for most combinations of AI-TC, except for grid cells with an AI value between 2 and 4 and a TC between 60 and 80, the calculated ratios were consistently greater than 1. This underscores a notable increase in correlation with TWS during the dry season compared to SSM.

Upon visualizing the results through the use of a heatmap, we noted discernible patterns, especially along the TC axis. Notably, as TC values increase, the observed ratio, r(NIRv~TWS_dry) - r( NIRv~TWS_growing) to r(NIRv~SSM_dry) - r(NIRv~SSM_growing), tends to decrease. It is important to acknowledge that this observation may stem from the higher pre-existing correlations with TWS during the growing season, leading to relatively smaller correlation increments during the dry months. We recognize that the primary point we intended to convey with the figure, emphasizing that correlation increases with TWS are more substantial than the correlation increases with SSM during dry months compared to growing season months, can already be effectively illustrated by Figure 3c. Therefore, we have chosen not to include the aforementioned figure to avoid redundancy.

**Figure 3 legend: This legend should stand alone and not require referencing Figure 2**

We will add the full legend to figure 3.

**Figure 4c: Can the authors explain the unusual pattern in the SHAP values at near-zero tree cover? I suspect this is a statistical artifact, but the high point density suggests it may be influencing results rather strongly.**

A zero tree cover fraction indicates the absence of trees in the grid cells. However, it's important to note that these areas may be covered by other types of short vegetation, such as grasslands or shrublands, or even partly non-vegetated land. The response of these alternative vegetation types to Soil Surface Moisture (SSM) and Total Water Storage (TWS) might vary, potentially accounting for the wide range and unusual patterns observed.

**Summary and Conclusions:  This section would benefit from stronger linkages to the existing site-level literature focused on vegetation water uptake depth, which is quite rich.**

We will link our results to existing site-level literature.

We will add that the stronger correlation of NIRv with SSM than TWSA is supported by site-level studies that find a higher root water uptake of surface soil moisture (Brinkmann et al., 2019, Gessler et al., 2021, Deseano Diaz et al., 2023; Kulmatiski and Beard, 2013), also when deeper water is available. Some local studies however find a higher root water uptake from deeper layers (Zhu et al., 2022).

In the next paragraph, we will add that local studies also found a different water uptake depth for trees and grasses in for example savanna ecosystems (Kulmatiski et al., 2010), and a different water uptake depth for tree species (Kahmen et al., 2022). Liu et al. (2021) showed for example that for a karst forest in Southwest China, evergreen species rely mostly on water sources from the 0-30 cm layer, while deciduous species extracted most water from the 30-70 cm layer.

Our global results are supported by site-scale studies that find that, during drought, the deeper roots play a more active role in water extraction (Stahl et al., 2013, Volkmann et al., 2016; Tao et al., 2021). In some studies however, the increase of deep water uptake is only relative: the absolute uptake of deep water does not increase, but the uptake of shallow water decreases (Brinkmann et al., 2019, Gessler et al., 2021, Rasmussen et al., 2020; Kühnhammer et al., 2023). This means that the uptake of deeper soil layers cannot compensate for the loss of water uptake from the dry topsoil. Contrary to trees, grasses do

not shift their uptake depth (Deseano Diaz et al., 2023), or even extract water from the most shallow soils (Prechsl et al., 2015, Kulmatiski and Beard, 2013).

To section 3.3, we will add that local studies identified other relevant factors that determine root water uptake depth such as forest stand age and tree height, competition, root hydraulic architecture, and tree species (Zhu et al., 2022; Quijano et al., 2012; Stahl et al., 2013, Gessler et al., 2021; Liu et al., 2021). For example, young trees more easily increase their root activity in the shallow or deep soil dependent on soil moisture than mature trees (Zhu et al., 2022; Drake et al., 2011). These variables were not included in our attribution analysis, because they are not available at global scale.

*Technical corrections:*

**Line 59 correct "allows to infer"**

**Line 65 correct typo near reference**

**Line 151 capitalize Spearman**

**Line 218-219 correct to "correlates more strongly with"**

**Line 253 correct Feldman citation format**

**Line 350 correct to "continuous near-surface  water availability"**

We thank the reviewer for pointing those out and we will correct them in the manuscript.

**Citations**

Chen, J., Liu, X., Ma, Y., Liu, L. (2022). Effects of Low Temperature on the Relationship between Solar-Induced Chlorophyll Fluorescence and Gross Primary Productivity across Different Plant Function Types. *Remote Sens. 14*, 3716. https://doi.org/10.3390/rs14153716

Zhu, W., Zhou, O., Sun, Y., Li, X., Di, N., Li, D., Yilihamu, G., Wang, Y., Fu, J., Xi, B., and Jia, L. (2023). Effects of stand age and structure on root distribution and root water uptake in fast-growing poplar plantations. Journal of Hydrology 616, 128831. https://doi.org/10.1016/j.jhydrol.2022.128831

Kulmatiski, A., Beard, K.H., Verweij, R.J.T., February, E.C. (2010). A depth-controlled tracer technique measures vertical, horizontal and temporal patterns of water use by trees and grasses in a subtropical savanna. New Phytologist 188(1), 199-209. https://doi.org/10.1111/j.1469-8137.2010.03338.x

Kulmatiski, A., and Beard. K.H. (2013). Root niche partitioning among grasses, saplings, and trees measured using a tracer technique. Oecologia 171, 25-37. https://doi.org/10.1007/s00442-012-2390-0

Tao, Z., Neil, E., Si, B. (2021). Determining deep root water uptake patterns with tree age in the Chinese loess area. Agricultural Water Management 249, 106810. https://doi.org/10.1016/j.agwat.2021.106810

Kühnhammer K., van Haren J., Kübert A., Bailey K., Dubbert M., Hu J., Ladd S.N., Meredith L.K., Werner C., Beyer M. (2023). Deep roots mitigate drought impacts on tropical trees despite limited quantitative contribution to transpiration. Science of the Total Environment 893, 164763. https://doi.org/10.1016/j.scitotenv.2023.164763

Prechsl, U.E., Burri, S., Gilgen, A.K., Kahmen, A., Buchmann, N. (2015). No shift to a deeper water uptake depth in response to summer drought of two lowland and sub-alpine C3-grasslands in Switzerland. Oecologia 177(1), 97-111. https://doi.org/10.1007/s00442-014-3092-6

Quijano, J.C., Kumar, P., Drewery, D.T., Goldstein, A., and Misson, L. (2012). Water Resources Research 48 (5). https://doi.org/10.1029/2011WR011416

Stahl, C., Hérault, B., Rossi, V., Burban, B., Bréchet, C., and Bonal, D. (2013). Depth of soil water uptake by tropical rainforest trees during dry periods: Does tree dimension matter? Oecologia 173 (4), 1191 - 1201. https://doi.org/10.1007/s00442-013-2724-6.

Brinkmann, N., Eugster, W., Buchmann, N., Kahmen, A. (2018). Species-specific differences in water uptake depth of mature temperate trees vary with water availability in the soil. Plant Biology 21 (1), 71-81. https://doi.org/10.1111/plb.12907

Deseano Diaz, P.A., Van Dusschoten, D., Kübert, A., Brüggemann, N., Javaux, M., Merz, S., Vanderborght, J., Vereecken, H., Dubbert, M. (2023). Response of a grassland species to dry environmental conditions from water stable isotopic monitoring: no evident shift in root water uptake to wetter soil layers. Plant and Soil 482 (1.2), 491 - 512. https://doi.org/10.1007/s11104-022-05703-y

Gessler, A., Bächli, L., Rouholahnejad Freund, E., Treydte, K., Schaub, M., Haeni, M., Weiler, M., Seeger, S., Marshall, J., Hug, C., Zweifel, R., Hagedorn, F. (2022). Drought reduces water uptake in beech from the drying topsoil, but no compensatory uptake occurs from deeper soil layers. New Phytologist 233 (1), 194-206. https://doi.org/10.1111/nph.17767

Volkmann, T.H.M., Haberer, K., Gessler, A., Weiler, M. (2016). High-resolution isotope measurements resolve rapid ecohydrological dynamics at the soil-plant interface. New Phytologist 210(3), 839-849. https://doi.org/10.1111/nph.13868

Rasmussen, C.R., Thorup-Kristensen, K., Dresboll, D.B. (2020). Uptake of subsoil water below 2 m fails to alleviate drought response in deep-rooted Chicory (Cichorium intybus L.). Plant and Soil 446(1-2), 275-290. https://doi.org/10.1007/s11104-019-04349-7

Kahmen, A., Basler, D., Hoch, G., Link, R.M., Schuldt, B., Zahnd, C., and Arend, M. (2022). Root water uptake depth determines the hydraulic vulnerability of temperate European tree species during the extreme 2018 drought. Plant Biology 24 (7), 1224-1239. https://doi.org/10.1111/plb.13476

Drake, P.L., Froend, R.H., Franks, P.J. (2011). Linking hydraulic conductivity and photosynthesis to water-source partitioning in trees versus seedlings. Tree Physiology 31 (7): 736-773. https://doi.org/10.1093/treephys/tpr068

---

## Referee Report (RR1)

*The authors have done a good job responding to the reviewer comments, including adding substantial additional analyses, appropriate limitations statements, and stronger links to the existing literature. The additional analyses reinforce the previous findings and provide new insights to the data. I have only a few minor remaining comments (in italics), which I have added to my initial comments (in bold) and the authors' response below (in plain text). After these have been addressed, I believe the manuscript is suitable for publication and will make an excellent contribution to the literature.*

**Lines 77-79: Assumptions 2 and 3 should be further justified and referenced. Assumption 2 is particularly concerning in grid cells with highly manipulated hydrologic systems, such as where irrigation results in substantial drawdown of groundwater and reservoir levels throughout the growing season.**

We will update this in the text in the same line with justifications.

*Perhaps I am missing it, but I do not see any text changes in response to this comment. Could you please 1) add a citation to support the assumption that canopy water storage is much smaller than soil water storage (across all biomes), and 2) add further justification, a citation, and/or a caveat pertaining to the assumption that water storage in lakes (including reservoirs) and groundwater is negligible? Short-term fluctuations in reservoir and volumes can be very large in some areas due to human management of water, as was recently comprehensively quantified by Cooley et al. Similarly, seasonal and sub-seasonal changes in groundwater storage can be very large in areas with intensive groundwater irrigation; for example, the papers by Strassberg et al. and Breña-Naranjo et al. listed below report short-term changes in groundwater storage in the High Plains Aquifer. I understand if this assumption is not avoidable, but it should acknowledged as a limitation and appropriately referenced.*

*Cooley, S. W., Ryan, J. C., & Smith, L. C. (2021). Human alteration of global surface water storage variability. Nature, 591(7848), 78-81.*

*Breña-Naranjo, J. A., Kendall, A. D., & Hyndman, D. W. (2014). Improved methods for satellite-based groundwater storage estimates: A decade of monitoring the high plains aquifer from space and ground observations. Geophysical Research Letters, 41(17), 6167-6173.*

*Strassberg, G., Scanlon, B. R., & Rodell, M. (2007). Comparison of seasonal terrestrial water storage variations from GRACE with groundwater-level measurements from the High Plains Aquifer (USA). Geophysical Research Letters, 34(14).*

**Figure 1c legend: Please specify what the white areas represent. Also, the color scale is variously described as blue and purple for positive correlations and red and orange for negative correlations in this legend and the legend for Figure S2; this should be standardized.**

We will specify in the caption that the white areas represent regions with no or insufficient number of data. Apart from this, the references to the figure's colors will be standardized.

*The caption is clearer, but the description of the legend colors is still inconsistent between Figs 1 and S2.*

**Line 238 and Figure S7: This figure is referenced in text after Figure S2; the SI figures should be reordered to be sequential.**

We will take it into consideration and update our manuscript accordingly.

*Figures S7 and S8 still appear to be out of order.*

*Finally, a number of typos remain (mostly related to spacing and punctuation). I did not record them all, but examples are present in lines 119, 127, 199, 325, 369, 393, 404 and 405, etc.*

---

## Author Response (AR2)

Reviewer 1

**The authors did a fantastic job responding to my comments. I really enjoyed reading this paper (acknowledging I am biased because of my research interests). Please consider a few minor comments below before publication. Well done.**

We thank the reviewer for his positive feedback on our manuscript and thank him for the constructive comments to improve our manuscript.

1) **I think it can be stated earlier that water limitation can reduce evaluation of source of soil water for roots. Energy-limitation will result in low correlations between any soil column water availability and NIRv. However, there might be differences in use of TWS and soil moisture for the plant water use though it can't be detected because of the lack of water limitation. For example, this can be mentioned earlier in line 170-172 about why some regions have negative correlations. This could partially be due to energy limitation.**

> In line 181 of the revised manuscript, we added:
>
> *Note that predominant energy limitation of the vegetation prevents the evaluation of the relevance of soil moisture vs. terrestrial water storage as partial correlations will become insignificant when temperature or radiation are mainly controlling vegetation functioning.*

2) **I think it is promising that use of SIF at 0.5 degrees shows similar results as shown in the SI. However, I do think it is worth at least mentioning some redundance and double counting of relationships that will be show in the boxes in Figs. 2 and 3. Specifically, GRACE is gridded at 0.5 degrees and really at 2 degrees. Matching 0.05 resolution NIRv from many pixels to the same GRACE anomalies can present large spatial autocorrelation to the results. I suggest mentioning this issue when mentioning the SIF results in the robustness section.**

> In last paragraph of the section 3.4, we added the following paragraphs:
>
> *When analyzing partial correlations between Total Water Storage (TWS) and vegetation metrics (NIRv or SIF) at finer resolutions (0.05 degrees for NIRv or 0.5 degrees for SIF), it is crucial to acknowledge the potential emergence of significant spatial autocorrelation. This is attributed to the fact that the actual spatial resolution of the satellite signal underlying the TWS data is 2-3 degrees.*

3) **I am satisfied with the bootstrapping on Fig. 2c. This does show a level of significant differences through aggregation. Additionally, consider moving Fig. S9 to the main text somewhere (I find this to be great). Or consider showing stippling on Fig. 1C which would require bootstrapping tests within individual pixels to see whether the correlation distribution for TWS-NIRv falls outside the bounds of the correlation distribution for soil moisture-NIRv. I leave this up to the authors.**

> *Following these suggestions, we utilized the bootstrapping technique to infer the significance of the results in the map in Figure 1c. Accordingly, a supplementary figure S4 has been included, illustrating that, for the majority of grid cells, the correlation distribution of NIRv ~ TWS falls outside the bounds of the correlation distribution for NIRv ~ SSM, indicating that the correlation difference displayed in Figure 1c is significant in many regions. Subsequently, Figure 1 has been revised to show the bootstrapping mean partial correlation which is supposed to be more robust than the actual partial correlations displayed before.*
>
> To convey this information, we incorporated the following details into line 167 in the Methodology section (2.2.2).

*To derive partial correlation estimates between NIRv and the water storages, we employed a bootstrapping approach (resampling with replacement from the original data) within each grid cell, with 1000 repetitions to compute bootstrap means and confidence intervals.*

In the caption of Figure 1, we included the following line:

*The color bar denotes the mean partial correlation for each grid cells, computed from the partial correlations across individual bootstrapping samples.*

Further in the line 269, we added:

*To ensure that the observed patterns of difference of partial correlation between SSM and TWS are not the artifacts arising from the computation of differences based on mean partial correlation, we compared the 95% confidence intervals obtained through bootstrapping. Our results indicate that, for the majority of the considered grid cells, the entire confidence intervals of the correlation (NIRv ~ TWS) fall outside the bounds of the correlation (NIRv ~ SSM) which indicates that the correlations differences are significant, thus enhancing the robustness and confidence in our findings (Figure S4).*

***Please refer to the line number from the revised manuscript.***

**Line 64: I suggest tempering slight with "providing valuable insights into at least some of the root zone soil moisture"**

The corresponding line is changed accordingly.

**Line 79: Optional, but could add this reference as well:**

Rodell, M. and Famiglietti, J. S.: An analysis of terrestrial water storage variations in Illinois with implications for the Gravity Recovery and Climate Experiment (GRACE), Water Resour. Res., 37(5), 1327–1339, doi:10.1029/2000WR900306, 2001.

We added the reference to the line 87.

*While soil moisture fluctuations represent the largest variation of TWS (Rodell and Famiglietti, 2001), it is essential to note that certain regions exhibit notable short term fluctuations in lake and groundwater due to human management (Strassberg et al., 2007; Cooley et al., 2021).*

**Line 302: "not always the case" (sometimes true)**

We change the line (line 326 in the revised manuscript):

This, however, is probably not the case and simply a reflection of reduced variability in surface soil moisture**.**

to

*This, however, is probably not the case and might simply be a reflection of reduced variability in surface soil moisture.*

**Line 344-359: References to figure 4 panels here appear incorrect**

Corrected within the paragraph.

Reviewer 2

**The authors have done a good job responding to the reviewer comments, including adding substantial additional analyses, appropriate limitations statements, and stronger links to the existing literature. The additional analyses reinforce the previous findings and provide new insights to the data. I have only a few minor remaining comments, which I have added to my initial comments and the authors' initial response below. After these have been addressed, I believe the manuscript is suitable for publication and will make an excellent contribution to the literature.**

We appreciate the positive feedback and constructive comments from the reviewer on our manuscript. Thank you for your valuable input in helping us improve our work.

**Initial reviewer comment:**

**Lines 77-79: Assumptions 2 and 3 should be further justified and referenced. Assumption 2 is particularly concerning in grid cells with highly manipulated hydrologic systems, such as where irrigation results in substantial drawdown of groundwater and reservoir levels throughout the growing season.**

**Author response:**

**We will update this in the text in the same line with justifications.**

**Follow up reviewer comment:**

**Perhaps I am missing it, but I do not see any text changes in response to this comment. Could you please 1) add a citation to support the assumption that canopy water storage is much smaller than soil water storage (across all biomes), and 2) add further justification, a citation, and/or a caveat pertaining to the assumption that water storage in lakes (including reservoirs) and groundwater is negligible? Short-term fluctuations in reservoir and volumes can be very large in some areas due to human management of water, as was recently comprehensively quantified by Cooley et al. Similarly, seasonal and sub-seasonal changes in groundwater storage can be very large in areas with intensive groundwater irrigation; for example, the papers by Strassberg et al. and Breña-Naranjo et al. listed below report short-term changes in groundwater storage in the High Plains Aquifer. I understand if this assumption is not avoidable, but it should acknowledged as a limitation and appropriately referenced.**

**Cooley, S. W., Ryan, J. C., & Smith, L. C. (2021). Human alteration of global surface water storage variability. Nature, 591(7848), 78-81.**

**Breña-Naranjo, J. A., Kendall, A. D., & Hyndman, D. W. (2014). Improved methods for satellite-based groundwater storage estimates: A decade of monitoring the high plains aquifer from space and ground observations. Geophysical Research Letters, 41(17), 6167-6173.**

**Strassberg, G., Scanlon, B. R., & Rodell, M. (2007). Comparison of seasonal terrestrial water storage variations from GRACE with groundwater-level measurements from the High Plains Aquifer (USA). Geophysical Research Letters, 34(14).**

We apologize that the previous comment of the reviewer slipped through. We agree with the reviewer and have added references (Cheng & Jia 2019, Stocker et al. 2023)to assumption 3 on negligible canopy water storage, and have emphasized the limitations associated with assumption 2, as suggested by the reviewer in line 85:

*While soil moisture fluctuations represent the largest variation in Total Water Storage (TWS) (Rodell and Famiglietti, 2001), it is crucial to note that certain regions exhibit significant short-term fluctuations in lake and groundwater due to human management (Strassberg et al., 2007; Cooley et al., 2021).*

**Initial reviewer comment:**

**Figure 1c legend: Please specify what the white areas represent. Also, the color scale is variously described as blue and purple for positive correlations and red and orange for negative correlations in this legend and the legend for Figure S2; this should be standardized.**

**Author response:**

**We will specify in the caption that the white areas represent regions with no or insufficient number of data. Apart from this, the references to the figure's colors will be standardized.**

**Follow-up reviewer comment:**

**The caption is clearer, but the description of the legend colors is still inconsistent between Figs 1 and S2.**

We thank the reviewer for catching this.

To make it more consistent,

We replaced "*The purple colour indicates that the correlation is positive while the red colour indicates a negative correlation between SSM and TWS*" in the caption of figure S2 with

"*The purple colour indicates the positive correlation of SSM with TWS while orange colour indicates the opposite.*"

**Initial reviewer comment:**

**Line 238 and Figure S7: This figure is referenced in text after Figure S2; the SI figures should be reordered to be sequential.**

**Author response:**

**We will take it into consideration and update our manuscript accordingly.**

**Follow-up reviewer comment:**

**Figures S7 and S8 still appear to be out of order.**

We have changed the order of Figures in Supplementary section, matching it with the order in which it appears in the text of the manuscript.

**Additional reviewer comment:**

**Finally, a number of typos remain (mostly related to spacing and punctuation). I did not record them all, but examples are present in lines 119, 127, 199, 325, 369, 393, 404 and 405, etc.**

We tried to removed most of the typos error in the updated manuscript.